# Effects of Iron Supplementation on Metabolism in Calves Receiving Whole Milk

**DOI:** 10.3390/ani13030477

**Published:** 2023-01-30

**Authors:** Anna Budny-Walczak, Kinga Śpitalniak-Bajerska, Marek Szołtysik, Krystyna Pogoda-Sewerniak, Robert Kupczyński

**Affiliations:** 1Department of Environment Hygiene and Animal Welfare, The Faculty of Biology and Animal Science, Wrocław University of Environmental and Life Sciences, 38c Chełmońskiego St., 50-375 Wrocław, Poland; 2Department of Functional Food Product Development, The Faculty of Biotechnology and Food Science, Wrocław University of Environmental and Life Sciences, 25 Norwida St., 50-375 Wrocław, Poland

**Keywords:** milk, calves, chelate, iron management, hematology, TIBC, UIBC, transferrin

## Abstract

**Simple Summary:**

Young animals in the early stages of life are most susceptible to iron deficiency; however, neonates do possess a certain degree of iron reserves in their body. Moreover, in the case of feeding calves with cow’s milk, which has a low iron concentration, rapid growth rates can lead to the development of temporary iron deficiency. This condition may be exacerbated by the immaturity of molecular mechanisms in relation to iron absorption. Further, iron supplementation is based primarily on inorganic compounds. However, these compounds can undergo oxidation and transform into insoluble forms. For the purposes of animal nutrition and to increase the bioavailability of this element, this research focuses on the use of chelates or proteinaceous iron preparations. Casein proteins possess excellent iron-binding properties, thereby decreasing their susceptibility to oxidation and therefore possessing high bioavailability.

**Abstract:**

The objective of this study was to determine the effects of feeding a protein–iron complex (PIC) to calves. Specifically, the aim was to understand how it influences productive performance and indicators of iron metabolism, hematology and biochemical and parameters during feeding with whole milk before weaning. The study was carried out on 20 Polish Holstein Friesian calves. The calves were then divided into a control group (CON), fed with full milk (n = 10), and an experimental group (MFe), who received a PIC additive in milk at 16 g/day (n = 10). In order to determine the production parameters, the calves were weighed at the beginning (i.e., on the 7th day of life) and at the end of the experiment (42nd day of life) using an electronic platform scale. Production parameters such as average weight gain (AWG), feed conversion ratio (FCR), and growth rate (GR) were assessed. Blood was collected from an external jugular vein (*vena jugularis externa*) on the 7th, 14th, 28th, and 42nd days of life. The mean daily gains in body weight (ADG), growth rate (GR), and the feed conversion ratio were highest in the experimental group, MFe. Therefore, it can be concluded that the addition of a protein–iron complex entailed a significant impact on the iron metabolism indicators in the MFe experimental group.

## 1. Introduction

The proper nutrition of calves in the neonatal period and in the first weeks of life has a fundamental impact on their growth, development, and later exploitation. The first liquid feed is either whole milk or a milk replacer. Milk replacers are commonly used due to their low cost, ease of storage and use, and positive impact on production rates [1,2]. With the low price of milk, breeders are extending the use of whole milk in calf nutrition more often. Whole milk is also used to feed calves on small organic farms. As the first liquid feed for calves, milk is a high-value source of protein, bioactive ingredients, and hormones, but is also low in iron. Calves—especially those raised for meat—fed exclusively with whole milk are thus exposed to deficiencies of this mineral component, which is concerning, as the mineral is responsible for the proper functioning of the immune system.

In recent years, the subject of research in relation to calf nutrition has been regarding the optimization of the sources of micro- and macro-elements, both in terms of their form and bioavailability. The demand coverage for minerals is important in terms of animal production, due to the fact that the deficiency of individual elements can cause a number of disturbances, thereby leading to a decrease in productivity [3,4]. Due to the importance of bioavailability, the form of compounds is also the subject of this research. Mohanta and Garg [5] proved that the absorption and use of trace elements are higher than those found in organic compounds. Furthermore, they also possess a positive effect on the oxidative and immune status, with a moderate effect on glucose and fatty acid metabolism [3,6]. In the case of milk replacers, increasing their properties can be achieved by using a number of functional additives, which include prebiotics, probiotics, non-protein nitrogen compounds, exogenous amino acids, nucleotides, lactoferrin, sodium butyrate, organic acids, chelates, or selected herbs [7,8,9,10,11,12,13,14]. Among the mushrooms with a probiotic effect, the most frequently mentioned are *Saccharomyces cerevisiae*, which produce a prosthesis with the ability to inactivate bacterial toxins and seal the intestinal barrier, thus acting as an antidiarrheal [15,16]. A less common species of yeast fungi is *Yarrowia lipolytica*. Strains of *Y. lipolytica* have been used as a feed additive for piglets [17] and turkeys [18]. In addition, these studies have shown a positive effect on animal growth.

The use of chelates as carriers of essential macro- and micro-nutrients has been the subject of research in various animal species [19,20,21]. Using additives containing iron compounds, bioavailability can be increased by chelation, which consists of a double covalent bond of the metal with amino acids. The administration of iron to calves is justified by the improvement of hematological indices and higher daily gains [22,23]. In this study, a preparation—in the form of an iron chelate based on casein—was developed with the participation of an enzyme that was isolated from a strain of *Yarrowia lipolytica*. The objective of this study was to determine the effects of feeding this protein–iron complex (PIC) to calves, specifically in relation to productive performance, as well as with respect to indicators of iron metabolism, hematology and biochemical parameters during feeding with whole milk before weaning.

## 2. Materials and Methods

### 2.1. Animals and Treatments

The study was carried on 20 Polish Holstein Friesian calves of the black-and-white variety. The study protocol was approved by the 2nd Local Bioethics Committee in Wroclaw (decision no. 63/2013).

The animals were put into randomized groups, taking into account age (7 days of age), body weight (ca. 40 ± 1.65 kg), and sex (50% females and 50% males in each group). Calves received full milk and granulated feed (Cargill calf starter, Poland) ad libitum. Clean water was available at all times. Milk was fed to the calves at 7:00 and 14:30 daily. During the experimental period, the calves received 6 l/head/day of milk. The calves had access to clean, fresh water, hay, and starter ad libitum.

The calves were divided into a control group (CON), fed with full milk r (*n* = 10), and an experimental group (MFe), who received a PIC additive in milk at 16 g/head/day (*n* = 10).

The body mass of the calves was measured before morning feeding on the 7th, 14th, 28th, and 35th day of age. Dry matter intake was controlled in this study.

During the study, calves were subjected to clinical observations. The viability, degree of dehydration, and fecal consistency were determined from a clinical trial and observations, which were conducted on the 7th, 14th, 28th, and 42nd days of life.

The chemical analysis of whole milk was performed at the Laboratory of Milk Assessment and Analysis at the Institute of Animal Husbandry and Breeding of the Wrocław University of Environmental and Life Sciences. The milk composition analysis was performed on the Milko-Scan 133 B device by Foss. Through the use of this instrument, the following was determined: fat, protein, lactose, dry matter (DM), and non-fat dry matter (NFDM). The analysis of the chemical composition of PIC was carried out in the laboratory of the Department of Animal Nutrition and Feed Science at the University of Life Sciences in Wrocław, in accordance with the recommendations of AOAC [24].

In order to determine production parameters, calves were weighed at the beginning (on their 7th day of age) and at the end of the experiment (42nd day of age) on an electronic platform scale designed for animals and with an accuracy of 0.2 kg (Radwag WP/4). Production parameters, such as average weight gain (AWG), feed conversion ratio (FCR), and growth rate (GR) were assessed according to the following formulas:(1)FCR=F ÷ (Bb−Be)
(2)$$GR=\frac{\left(Bb-Be\right)}{0.5\times \left(Bb+Be\right)}\times 100\%$$
where

F—feed intake (kg);Bb—beginning body weight (kg);Be—ending body weight (kg).

### 2.2. Blood Collection and Analysis

Blood was collected from the calves’ external jugular vein (*vena jugularis externa*) on the 7th, 14th, 28th, and 42nd days of age. The 7th day of age was taken as the start of the experiment (with regard to PIC supplementation). Blood was collected into a tube without anticoagulant, as well as into one containing K2EDTA (Sarstedt, Warszawa, Poland). The blood samples for serum and plasma were centrifuged at 3000× *g* for 10 min at room temperature (2 h after collection). Serum samples were then frozen (−20 °C) until the analysis.

Blood laboratory tests were performed in order to assess the influence of used preparations on hematological, biochemical, and immunological parameters. Hematological parameter analyses were performed using an ABC Vet analyzer (Horiba ABX, France). In conducting these analyses, certain parameters were taken into account, such as: red blood cells (RBCs); white blood cells (WBCs); platelets (PLTs); hemoglobin (HGB); hematocrit (HCT); mean corpuscular volume (MCV); mean corpuscular hemoglobin (MCH); mean corpuscular hemoglobin concentration (MCHC); lymphocytes (LYMs); and monocytes (MONs).

The iron (Fe) was infused into the serum via a photometric test using a Pentra 400 Horiba ABX biochemical analyzer (France).

The unsaturated iron-binding capacity (UIBC) was determined in the serum by using the ferrozine photometric method, also using Pointe Scientific reagents.

The total iron-binding capacity (TIBC) was determined in the serum by the photometric method with the precipitation of Fe^+3^ alkaline calcium carbonate. This was achieved by using a BioMaxima precipitation reagent, followed by a Pentra 400 Horiba ABX analyzer (France) iron determination reagent. The transferrin in serum was determined with the bovine transferrin ELISA Kit, Bethyl company. The transferrin saturation with iron (TS) was calculated according to the following formula:(3)$$TfS\;\left(\%\right)=\left[Fe\left(\frac{µ\mathrm{moL}}{L}\right)\right]\;÷\;TIBC\left(\frac{µ\mathrm{moL}}{L}\right)\times 100$$

Laboratory analyses with respect of the blood serum were performed using a Pentra 400 analyzer (Horiba ABX, France). The following parameters were estimated:Glucose, via the oxidase method; the reagents used were obtained from HORIBA ABX (France);Blood biochemical parameters, i.e., glucose, free fatty acids (NEFA), β-hydroxybutyric acid (BHBA), total protein (TP), albumin (Alb), lactic acid (LA), and cholesterol (Col.);Enzymes, i.e., aspartate aminotransferase (AST), alanine aminotransferase (ALT), and γ-glutamyltransferase (GGT).

In addition, the liver enzymes were measured according to IFCC recommendations.

### 2.3. Process of Obtaining the Protein–Iron Complex

Process of obtaining casein complex–PIC was thoroughly described in manuscript Kupczyński et al. [25]. The iron contents were analyzed using an atomic absorption spectrophotometer Varian SpectrAA 220 (Agilent Technologies Inc. Palo Alto, Santa Clara, CA, USA) [26]. This analysis was used to determine the dose of PIC in treatment groups.

### 2.4. Statistical Analysis

Data were analyzed using a general linear model for repeated measures based on the MIXED procedure of SAS (version 9.2; SAS Institute Inc., Cary, NC, USA). For the purposes of repeated measures, the model included the effect of dietary treatments (D) and sampling time (T) as fixed effects and their interactions (D × T) were recorded according to the following model:(4)Yijk=µ+αi+βj+αβij+εijk
where

Yijk—dependent variable;µ—an overall mean;αi—dietary treatment effect (two groups);βj—series of blood tests;αβij—treatment effect x series of tests;εijk—random residual error.

The same statistical model was applied for the average daily weight gain (ADG), growth rate (GR), feed conversion factor (FCR), and the parameters of health status. These data were analyzed as repeated measures using a MIXED procedure of SAS with the measurement day as the repeated variable, which was used for the estimated time effect βj factor (1, 2, 3, …, *n*). Before the analyses, all data were screened for normality using the UNIVARIATE procedure of SAS. Mean values were compared with Duncan’s test. The data were presented as average values and accompanied by a standard error of the mean. Finally, significant differences were declared at *p* < 0.05.

## 3. Results

### 3.1. Animals and Treatments

The nutritional value of doses for individual experimental groups is presented in Table 1. The iron concentration in the whole milk was 3 mg/kg (group CON), while in the PIC formula the concentration was at set 407.05 mg/kg, which is a 4.09 mg Fe/kg concentration with respect to the feed ration in group MFe.

Overall, health scores values showed no statistically significant differences between the groups with respect to vitality, dehydration, and stool consistency over the entire study period. Average daily weight gain (ADG), growth rate (GR), and the feed conversion ratio (FCR) were highest in the experimental group MFe (Table 2). Furthermore, the ADG in the MFe group was 738.13 g/day, while in the CON group it was 556.50 g/day. The GR was higher in the experimental group by 6.18%, and in the FCR by approximately 18% when compared with the control group.

### 3.2. Blood Analysis

In both the CON and MFe groups, a gradual decrease in the number of leukocytes (WBC) was observed during the study period, which in the CON group was noted as being statistically significant (*p* < 0.01) (Table 3). The mean WBC was lower in the MFe group when compared with the CON group. The lowest mean values of HGB, HCT, and PLT were found in the calves from the control group. The RBC value, despite some changes during the study period, did not differ between the groups. There were statistically significant differences at *p* < 0.01 in the MCV concentration at the beginning and end of the experiment in both groups. In addition, the mean MCH concentration in the MFe group was statistically lower (*p* < 0.01) than in the CON group. During the experiment, the MCHC value increased in the control group (*p* < 0.01); however, with respect to the MFe group, the mean values were even (Table 3).

At the beginning of the experiment, the iron concentration in the blood was low in both groups (Table 4). During the study period, the iron concentration significantly increased in both groups. Moreover, the greater mean value at the end of the experiment was found in the MFe group. Further, the changes in the concentration of iron were accompanied by similar changes with respect to transferrin in the blood. In addition, the mean concentration of transferrin was statistically higher in the MFe group when compared with the CON group (*p* < 0.01).

Both the UIBC and TIBC concentrations increased during the study in both groups, with the highest TIBC concentration noted at the 42nd day of life in the CON group (TIBC: 47.24 μmol/L). In the MFe group, along with the increase in transferrin and transferrin saturation, TIBC also increased. However, these changes were not noted for transferrin in the CON group.

In the MFe group, a decrease in NEFA concentration was found at the end of the experiment (Table 5) when compared with the CON group (*p* < 0.05). The average concentration of BHBA in both groups increased during the experiment. On the day of completion for the study, the concentration of total cholesterol increased in the MFe group. It was noted that these changes were similar in the CON group as well. The decreasing trends in the lactic acid concentration were observed in both groups. In the CON group, a decrease in LA concentration (*p* < 0.05) was found between the start and end of the study. However, there were no differences in the concentration of glucose with respect to the blood serum of the calves between the groups.

The mean TP concentration was lower (*p* < 0.05) in the MFe group when compared with the CON group. In addition, the serum albumin concentrations increased during the study in both groups. Statistically, the albumin increase was noted in the CON calves (*p* < 0.05), while a higher mean concentration was noted in the MFe calves. In the case of AST activity, an increase in the activity of this enzyme was found in the control group, but the mean value for the entire study period did not differ between the groups. Moreover, the ALT activity showed a downward trend, while statistically significant differences (*p* < 0.05) were noted in the MFe group between the 7th and 42nd days of the calves’ life. The activity of γ-glutamyltransferase was in high concentration on the 7th day of the study, but it decreased during the rest of the experiment period. In the CON group, this decrease was found to be statistically significant (*p* < 0.05).

## 4. Discussion

Calves are at risk of anemia when they are exclusively fed a whole milk diet for the first few weeks of their life [27]. In addition, iron deficiency can adversely affect growth, immunity, and feed conversion. The recorded average daily weight gains were within the range of values considered normal for calves at this age (according to the NRC) [27], but the differences between the groups were not found to be statistically significant.

The calf’s Fe reservoir is mainly in the liver and spleen. Further, its reserves are generally sufficient to prevent severe anemia if the calves have access to solid feed [23]. In the case of slight disturbances in iron management, a sufficiently high-iron content in a milk replacer and solid feed renders it possible to compensate for the deficit over a short period of time. In the case of whole-milk feeding, iron stores are depleted in the first 3–4 weeks of life. Moreover, the clinical signs of Fe deficiency appear around two months of age and worsen when Fe deficiency cannot be compensated.

The decrease in RBCs during the first weeks of life after birth is a physiological phenomenon indicating the normal development of calves and is not related to the level of iron [6]. However, it should be noted that it could have been caused by the shorter lifespan of fetal erythrocytes [28]. Having said this, Mohri et al. [6] suggested that iron supplementation in the first month of life in calves prevents a decrease in RBCs. In the MFe group, an increase in RBCs was observed up to the 28th day of life; while in the control group CON, this parameter was even, which is comparable to other studies [6]. Calves receiving 10% iron dextran intramuscularly showed only a lower variability of red blood cell parameters, despite the lack of development of anemia in the control group [29]. Other authors report an increase in the concentration of hemoglobin and the number of erythrocytes in the blood when iron preparations are administered to calves [22,23], but this is achieved without affecting red blood cell indices [23].

Ježek et al. [30] describe the increase in WBCs after birth, which was observed to a small extent in the control group. However, a gradual decrease in the number of leukocytes (WBCs) was observed in the experimental group. A normal WBC concentration may indicate the absence of inflammation in both groups.

Values of MCV and MCH parameters in the experiment were noted to be below the normal values [31]. Indeed, Mohri et al. [23] also showed a decreasing trend for MCV and MCH. Additionally, Knowles et al. [32] reported that MCV decreased to the lower limit of the reference range. The size of erythrocytes decreases for the first 3–4 months after the neonatal period of calves [23]. A gradual decrease in MCV coincides with a disappearance of fetal hemoglobin and its replacement by hemoglobin A [23,33]. With a simultaneous decrease in RBC concentration, a decrease in MCV may indicate a correlation between average erythrocyte volume and hemoglobin production, regardless of the level of Fe in the blood serum [23]. However, Miltenburg et al. [34] reported that MCV correlated with iron administration. Moreover, it was only in the CON group that the MCHC value increased at the end of the experiment. In the MFe group, the MCHC also assumes an upward trend, which was also shown by Knowles et al. [32] from birth up to three months of age in calves.

In the case of feeding calves with whole milk, there is a large discrepancy in the concentration of iron in the blood serum (10.39–20.33 μmol/L) [32]. Furthermore, Knowles et al. [32] report a concentration of 15 μmol/L Fe in the blood of clinically healthy calves in the first week of life, while Mohri et al. advise a concentration of 19.22 μmol/L [23]. Primary iron deficiency may lead to a decrease in its concentration in the blood serum (<17 μmol/L), but may also result in an increase in TIBC and UIBC [35,36]. In our own research, the concentration of Fe in the MFe group was 24.10 μmol/L at the end of the study. Furthermore, the UIBC and TIBC mean values were lower in comparison to the CON group (UIBC 14.64 μmol/L and TIBC 32.72 μmol/L).

An iron-balanced diet is essential for the production of erythrocytes and hemoglobin [37]. Further, additional supplementation can play an important role in calf growth and also resistance to infection [38]. Mohri et al. [23] showed an increase in Fe concentration up to the 28th day of life (28.67 μmol/L) after the use of a supplement in the form of iron sulphate. A similar relationship was noted in the MFe group. In other studies, an increase in Fe concentration was observed up to the 60th day of life [39]. In monogastric animals, research shows that the efficiency of using iron chelate is almost twice as high as in the case of FeSO_4_ [40]. The use of complex compounds with respect to iron hydrolyzates has been confirmed in studies that were also conducted on model animals [41].

After oral iron supplementation was added to whole milk (i.e., the MFe group), a statistically significant increase in blood transferrin was found, which is similar to the results of other studies [23,42]. Tothova et al. [43] reported that the concentration of transferrin increases to 8 mg/mL in meat calves with an iron deficiency, constituting a negative correlation with hemoglobin concentration, which is not confirmed by our study. In the described study, an increase in transferrin up to 6.48 mg/mL was observed with no significant changes in hemoglobin concentration. Moreover, TIBC and UIBC showed an increasing trend up to the 28th day of life in both groups, while other reports showed a reverse trend up to the 21st day of life [42].

In the MFe group, a significant decrease in the concentration of β-hydroxybutyric acid was observed. Knowles et al. [32] indicated an increase in BHBA concentration between the 1st–5th and 10th–80th day of life. A higher concentration of BHBA in the first few weeks of life, when intake of solid feed was insignificant, may be the result of hypoglycemia, negative energy balance, and weight loss [44], which was not found in this study. The concentration of BHBA increases with the calves’ age [45] as a result of an increased intake of solid feed [45]. In addition, it is currently understood to be an indicator of rumen development [9,45,46]. Further, higher values for NEFA were noted by Knowles et al. [32], at similar concentrations of β-HM.

Cholesterol concentrations on day 28th and 42nd exceeded the reference values for calves in both groups [47]. The concentration increase in this parameter during the study period most likely resulted from a high supply of fat in whole milk. Indeed, glucose levels from the 14th day of life decreased with age in all calves, which may be related to elevated corticosteroid levels and the development of forestomachs [23]. On 14th day of life, the concentration exceeded the reference values [33]. Such a relationship is also indicated by Mohri et al. [23] and Knowles et al. [32], who reported higher glucose concentrations in the first few days of life. Moreover, insufficient glucose uptake was observed when the calves were fed with whole milk [48].

In the calves’ neonatal period, the TP concentration was directly affected by the amount and time of colostrum intake [34]. The addition of iron in the MFe group resulted in a decrease in TP concentration by day 42. Downward trends were also observed by Mohri et al. [23]. A lower concentration of TP was clearly noted by Prodanović et al. [26] with respect to calves with an iron deficiency; this may be related to IgG absorption in the neonatal period [49]. Similar correlations were noted in the studies of Kupczyński et al. [25,50], thereby indicating the negative impact of a higher amount of iron in the diet on IgG levels.

In this experiment, a higher concentration of albumin was noted on day 42 in the MFe group. A similar dependence of an Alb increase with a TP decrease is indicated by Mohri et al. [6] and Knowles et al. [32]. Furthermore, the concentration of TP and Alb is influenced by nutrition and liver function [51]. The increase in serum albumin concentration may be related to compensation with respect to the decreasing osmotic pressure in plasma, which is due to the low level of globulins [23].

It must be noted that AST is a sensitive indicator of liver damage, but other reports also describe elevated AST levels in calves up to 84 days of age [6]. Moreover, Klinkon and Ježek [51], as well as Mohri et al. [6], observed a decrease in AST by the third week of life, followed by a moderate increase. Similar relationships were noted for the MFe group. In addition, the measurement of AST activity in conjunction with CK was used to diagnose muscle damage [51]. Lastly, the supplementation of PIC used in this experiment resulted in no significant effect on the activity of tested enzymes.

## 5. Conclusions

The addition of a protein–iron complex (PIC) had a significant impact on iron metabolism indicators—thereby resulting in an increase in iron concentration in calves’ whole blood—as well as with respect to TIBC, UIBC, and transferrin concentrations in the MFe experimental group. The findings of this study on the efficacy or potential of a protein-mineral chelate suggest that it possesses functional properties that could be useful in the treatment/prevention of anemia in calves. Future studies should focus on confirming the results obtained in more animals.

## Figures and Tables

**Table 1 animals-13-00477-t001:** The nutritional value of doses for individual experimental groups in DM.

Item	Group	
	CON	MFe
DM [g/kg]	136.9	139.1
Gross energy [MJ/kg]	19.89	19.84
Ash [% DM]	0.70	0.70
Protein [% DM]	3.14	3.37
Fat [% DM]	5.09	5.08
Na [g/kg]	3.8	3.8
Cl [g/kg]	9.2	9.2
K [g/kg]	11.2	11.2
Fe [mg/kg]	3.00	4.09

CON—control group; MFe—experimental group; and DM—dry matter.

**Table 2 animals-13-00477-t002:** Average daily weight gain (ADG), feed conversion factor (FCR), and growth rate (GR).

Item	Group		SEM	*p*-Value
	CON	MFe		
ADG [g/day]	556.50	738.13	66.33	0.19
FCR [kg]	2.29	2.70	0.15	0.07
GR [%]	33.85	40.03	3.62	0.43

CON—control group; MFe—experimental group; ADG—average daily weight gain; FCR—feed conversion factor; and GR—growth rate.

**Table 3 animals-13-00477-t003:** Mean values of the hematological parameters with regard to the calves’ blood.

Day	Treatment		SEM	*p*-Value		
	CON	MFe		D	T	D × T
WBC [G/L]						
d 7.	12.13 ^A^	10.23	1.11	0.21		
d 14.	10.77 ^a^	8.44	1.26	0.04		
d 28.	8.58	7.87	0.80	0.70		
d 42.	6.00 ^bB^	8.55	0.59	0.03		
$\bar{x}$	9.37	8.77	0.53	0.09	0.04	0.35
RBC [G/L]						
d 7.	7.48	7.15	0.27	0.57		
d 14.	7.28	6.90	0.45	0.72		
d 28.	6.84	7.70	0.44	0.39		
d 42.	6.46	7.44	0.27	0.97		
$\bar{x}$	7.02	7.23	0.17	0.72	0.13	0.10
HGB [mmoL/L]						
d 7.	5.78	6.18	0.23	0.40		
d 14.	5.61	6.20	0.37	0.50		
d 28.	5.63	6.05	0.24	0.46		
d 42.	5.69	6.33	0.22	0.16		
$\bar{x}$	5.69	6.19	0.13	0.33	0.95	0.93
HCT [L/L]						
d 7.	0.30	0.29	0.01	0.56		
d 14.	0.28	0.27	0.02	0.70		
d 28.	0.25	0.28	0.02	0.44		
d 42.	0.26	0.27	0.01	0.76		
$\bar{x}$	0.27	0.28	0.01	0.77	0.03	0.06
PLT [G/L]						
d 7.	579.67	782.89	76.49	0.20		
d 14.	709.00	754.86	70.78	0.79		
d 28.	680.83	746.67	70.51	0.69		
d 42.	690.17	787.88	58.66	0.43		
$\bar{x}$	658.62	773.07	34.51	0.50	0.44	0.26
MCV [fL]						
d 7.	40.83 ^A^	40.44 ^A^	0.32	0.57		
d 14.	38.67	38.71	0.45	0.96		
d 28.	36.17	36.33	0.36	0.85		
d 42.	34.83 ^B^	35.50 ^B^	0.28	0.26		
$\bar{x}$	37.48	38.07	0.37	0.73	<0.01	0.63
MCH [fmoL/L]						
d 7.	0.83	0.80	0.01	0.27		
d 14.	0.85	0.81	0.01	0.003		
d 28.	0.90 **	0.73 **	0.04	0.02		
d 42.	0.85 *	0.76 *	0.02	0.01		
$\bar{x}$	0.86 **	0.79 **	0.01	0.11	0.23	0.12
MCHC [mmoL/L]						
d 7.	20.45 ^A^	19.94	0.25	0.35		
d 14.	22.17	21.21	0.33	0.21		
d 28.	24.87 ^B^**	20.40 **	1.06	0.03		
d 42.	24.58 ^B^*	21.46 *	0.64	0.01		
$\bar{x}$	23.14 **	20.77 **	0.34	0.22	0.02	0.07

ab and AB—statistical difference within a group (*p* < 0.05 and *p* < 0.01); * and **—statistical difference between the groups (*p* < 0.05 and *p* < 0.01). CON—control group; MFe—experimental group; WBC—white blood cell; RBC—red blood cell; HGB—hemoglobin; HCT—hematocrit; PLT—platelet count; MCV—mean corpuscular volume; MCH—mean corpuscular hemoglobin; and MCHC—mean corpuscular hemoglobin concentration.

**Table 4 animals-13-00477-t004:** Mean values of iron management parameters in the calves’ blood.

Day	Treatment		SEM	*p*-Value		
	CON	MFe		D	T	D × T
Fe [µmoL/L]						
d 7.	7.24 ^A^	10.18 ^A^	2.06	0.50		
d 14.	9.47	14.21	3.47	0.56		
d 28.	18.20 ^B^	25.08 ^B^	3.30	0.36		
d 42.	18.28 ^B^	24.10 ^B^	2.62	0.29		
$\bar{x}$	13.29	15.28	1.54	0.69	0.01	0.02
Transferrin [mg/mL]						
d 7.	2.37	3.32	0.94	0.32		
d 14.	3.49	3.98	1.07	0.85		
d 28.	2.86	6.48	2.65	0.005		
d 42.	2.95	5.13	0.35	0.01		
$\bar{x}$	2.92 **	4.73 **	0.42	0.003	0.02	0.07
UIBC [µmoL/L]						
d 7.	5.01 ^A^	8.26	1.76	0.39		
d 14.	7.46	8.42	1.72	0.83		
d 28.	16.25	16.49	3.52	0.98		
d 42.	17.07 ^B^	19.64	2.32	0.07		
$\bar{x}$	11.86	13.21	1.34	0.16	0.001	0.02
TIBC [µmoL/L]						
d 7.	18.34	12.15	3.66	0.43		
d 14.	22.56	15.16	4.53	0.53		
d 28.	41.26	33.31	6.95	0.62		
d 42.	47.24	32.72	4.77	0.14		
$\bar{x}$	22.35	23.34	2.78	0.60	<0.001	0.001
Transferrin saturation [%]						
d 7.	59.87	60.25	4.17	0.97		
d 14.	51.58	59.77	8.07	0.70		
d 28.	52.21	60.11	2.65	0.16		
d 42.	51.00	58.24	3.47	0.32		
$\bar{x}$	54.18	59.51	2.32	0.68	0.91	0.67

AB—statistical difference within a group (*p* < 0.01). **—statistical difference between the groups (*p* < 0.01). CON—control group; MFe—experimental group; UIBC—unsaturated iron binding capacity; and TIBC—total iron binding capacity.

**Table 5 animals-13-00477-t005:** Mean values of the selected biochemical in calves’ blood.

Day	Treatment		SEM	*p*-Value		
	CON	MFe		D	T	D × T
NEFA [mmoL/L]						
d 7.	0.28	0.28	0.02	0.86		
d 14.	0.21	0.22	0.02	0.86		
d 28.	0.22	0.32	0.04	0.92		
d 42.	0.33 *	0.16 *	0.03	0.004		
$\bar{x}$	0.27	0.23	0.02	0.30	0.82	0.78
BHBA [mmoL/L]						
d 7.	0.09	0.05	0.01	0.01		
d 14.	0.06	0.05	0.01	0.51		
d 28.	0.13	0.14	0.02	0.86		
d 42.	0.15	0.19	0.02	0.07		
$\bar{x}$	0.11	0.12	0.01	0.10	0.01	0.01
Chol. [mmoL/L]						
d 7.	1.56	1.91	2.16	0.22		
d 14.	1.28	1.93	0.30	0.34		
d 28.	2.23	3.11	0.28	0.15		
d 42.	6.28	5.39	3.37	0.19		
$\bar{x}$	3.06	3.09	1.30	0.55	0.24	0.39
LA [mmoL/L]						
d 7.	3.02 ^a^	3.33	0.47	0.76		
d 14.	2.44	2.34	0.48	0.95		
d 28.	0.95	1.33	0.10	0.05		
d 42.	0.77 ^b^	1.18	0.08	0.004		
$\bar{x}$	1.72	2.13	0.23	0.62	0.18	0.41
Glucose [mmoL/L]						
d 7.	5.82	6.25	0.32	0.53		
d 14.	7.33	6.58	0.53	0.54		
d 28.	6.38	6.54	0.35	0.85		
d 42.	5.93	5.29	0.28	0.28		
$\bar{x}$	6.23	6.08	0.19	0.98	0.28	0.68
TP [g/L]						
d 7.	58.87	57.17	5.20	0.12		
d 14.	54.30	51.36	2.16	0.56		
d 28.	55.05	50.53	1.13	0.05		
d 42.	47.16	49.25	6.79	0.13		
$\bar{x}$	53.78 *	52.08 *	2.76	0.86	0.24	0.55
Alb [g/L]						
d 7.	25.92 ^a^	30.08	2.23	0.38		
d 14.	26.40 ^a^	29.81	0.61	0.68		
d 28.	26.20	29.43	0.83	0.06		
d 42.	32.25 ^b^	42.68	4.04	0.21		
$\bar{x}$	27.88	33.01	1.53	0.43	0.26	0.38
AST [U/L]						
d 7.	34.73 ^A^	35.07	1.38	0.91		
d 14.	31.93 ^A^	38.67	2.37	0.21		
d 28.	39.50 ^a^	49.50	2.58	0.06		
d 42.	54.13 ^bB^	43.96	2.51	0.04		
$\bar{x}$	41.24	40.24	1.34	0.91	<0.001	0.09
ALT [U/L]						
d 7.	10.30	11.33 ^a^	0.69	0.48		
d 14.	9.40	8.51	0.42	0.36		
d 28.	7.78	7.20	0.58	0.67		
d 42.	10.35	8.11 ^b^	0.51	0.02		
$\bar{x}$	9.47	9.02	0.33	0.12	0.02	0.28
GGT [U/L]						
d 7.	105.68 ^a^	91.25	11.26	0.04		
d 14.	83.35	84.31	21.59	0.99		
d 28.	39.29	17.63	7.81	0.21		
d 42.	23.60 ^b^	22.33	2.56	0.81		
$\bar{x}$	57.79	53.88	7.15	0.73	<0.01	0.30

ab and AB—statistical difference within a group (*p* < 0.05 and *p* < 0.01). * —statistical difference between the groups (*p* < 0.05). CON—control group; MFe—experimental group; NEFA—non-esterified fatty acid; BHBA—beta-hydroxy-butyric acid; Chol.—cholesterol; LA—lactic acid; TP—total protein; Alb—albumin; AST—aspartate transaminase; ALT—alanine transaminase; and GGT—gamma-glutamyltransferase.

## Data Availability

Available from the corresponding authors.

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
