# Peer review of "Effects of Iron Supplementation on Metabolism in Calves Receiving Whole Milk"

_animals, 2023, doi:10.3390/ani13030477_

Round 1
Reviewer 1 Report
Thank you for inquiring about manuscript Manuscript ID: animals-2140071. I hope you find my comments helpful in making your decision about whether to publish the manuscript. Please find my additional comments attached.
- What is the main question addressed by the research? Is it relevant and
interesting? How original is the topic? What does it add to the subject
area compared with other published material?
The study investigated the potential use of protein-iron complex 21 (PIC) in calf diets prior to weaning to improve performance in Polish Holstein Friesian calves. In comparison to other published materials, the current manuscript demonstrated the potential role of protein-mineral chelate in the management of neonatal calf anaemia. The topic of interest in the field of calf nutrition, rearing, and management. When calf milk is supplemented with iron, minerals, and vitamins, young calves are known to absorb them very well. Nonetheless, the research is relevant and interesting and addressed an important topical issue related to the subject field.
- Is the paper well written?Is the text clear and easy to read? Are the conclusions consistent with
the evidence and arguments presented? Do they address the main question
posed?
Given the authors' findings and the logical presentation of the discussion, I find the content to be easy to read and consistent with the outcomes. The main question has been addressed, and the study design is satisfactory. However, given the study's sample size, I believe the conclusion should be reconsidered. A comment on the study's choice of such a small sample size could be useful in supporting the content of MS.
L 263: No such thing as a reference! The authors should double-check the referenced material.
L 339 – 344: Addition of a protein-iron complex PIC had a significant impact on the iron metabolism indicators and contributed to an increase in the concentration of iron in callves’ whole blood, as well as TIBC, UIBC and concentration of transferrin in experimental group MFe. Conducted evaluation of effectiveness of protein-mineral chelate indicates its functional features that should be used in practice as an anaemia prevention in calves.
The addition of a protein-iron complex PIC had a significant impact on iron metabolism indicators, resulting in an increase in iron concentration in calves' whole blood, as well as TIBC, UIBC, and transferrin concentration in the experimental group MFe. The findings of a study on the efficacy or potentials of protein-mineral chelate suggest that it has functional properties that could be useful in the treatment/prevention of anaemia in calves.
Re-write (p <0.01) as (p <0.01) all through the MS.
Author Response
Dear Reviewer,
Thank you for your valuable comments regarding manuscript ID: animals-2140071, entitled Effects of Iron Supplementation on Metabolism, Oxidative and Immune Status in Calves receiving whole milk. We appreciate your detailed review and hope that our statements will find your acceptance. English language editing by MDPI (certificate attached, changes shown in blue or red).
Thank you for inquiring about manuscript Manuscript ID: animals-2140071. I hope you find my comments helpful in making your decision about whether to publish the manuscript. Please find my additional comments attached.
-What is the main question addressed by the research? Is it relevant and
interesting? How original is the topic? What does it add to the subject
area compared with other published material?
The study investigated the potential use of protein-iron complex 21 (PIC) in calf diets prior to weaning to improve performance in Polish Holstein Friesian calves. In comparison to other published materials, the current manuscript demonstrated the potential role of protein-mineral chelate in the management of neonatal calf anaemia. The topic of interest in the field of calf nutrition, rearing, and management. When calf milk is supplemented with iron, minerals, and vitamins, young calves are known to absorb them very well. Nonetheless, the research is relevant and interesting and addressed an important topical issue related to the subject field.
AU: Authors conducted research on a new supplement containing iron. This supplement has been characterized previously. Thank you for recognizing the importance of our research.
- Is the paper well written? Is the text clear and easy to read? Are the conclusions consistent with the evidence and arguments presented? Do they address the main question posed?
Given the authors' findings and the logical presentation of the discussion, I find the content to be easy to read and consistent with the outcomes. The main question has been addressed, and the study design is satisfactory. However, given the study's sample size, I believe the conclusion should be reconsidered. A comment on the study's choice of such a small sample size could be useful in supporting the content of MS.
AU: The number of animals wasn’t large, but sufficient to statistically verify the results of our research. Added sentence at the end of conclusions: Line 383-384: Future studies should focus on confirming the results obtained in more animals.
L 263: No such thing as a reference! The authors should double-check the referenced material.
AU: In this part of the work, the discussion concerns work number 26 (Line 271 after corrections and 294): Ježek, J.; Nemec, M.; Staric, J.; Klinkon, M. Age related changes and reference intervals of haematological variables in dairy calves. Bull Vet Inst Pulawy 2011, 55, 471–478.
“Our results regarding the WBC count are in accordance with findings of Knowles et al. (11), who studied 14 crossbred calves and demonstrated only a higher number of WBC count after birth, which decreased later. Similar dynamics in calves until the age of 6 months was established in Norway. Number of WBC ranged between 9.4 and 12.0 x 109/L and were in the range of reference values for adult animals (2). Egli and Blum (5) found that the WBC number decreased from birth to the 42nd d, then it increased slightly until the 84th d. Terosky et al. (20) reported lower WBC number comparing to our results. Calves in their study received milk replacer and starter with antibiotics to the 5th week, which could be the reason for lower WBC number.”
Jezek et al. indicated independent fluctuations of WBC, however, the trend line in the graph shows a slight increase in the parameter up to 4/5 weeks of age. Jezek et al. in their discussion also invoke research of Knowles et al. (2000), which showed a similar relationship. The posted sentence has been changed to: “Ježek et al. [26] describe the increase of WBC after birth, which was observed to a small extent in control group.” Line 294-295.
L 339 – 344: Addition of a protein-iron complex PIC had a significant impact on the iron metabolism indicators and contributed to an increase in the concentration of iron in callves’ whole blood, as well as TIBC, UIBC and concentration of transferrin in experimental group MFe. Conducted evaluation of effectiveness of protein-mineral chelate indicates its functional features that should be used in practice as an anaemia prevention in calves.
- Done according to the Reviewer's suggestion (Line 378-383). The addition of a protein-iron complex PIC had a significant impact on iron metabolism indicators, resulting in an increase in iron concentration in calves' whole blood, as well as TIBC, UIBC and transferrin concentration in the experimental group MFe. The findings of a study on the efficacy or potentials of protein-mineral chelate suggest that it has functional properties that could be useful in the treatment/prevention of anaemia in calves.
Re-write (p <0.01) as (p <0.01) all through the MS.
AU: Done according to the Reviewer's suggestion (full text).

Reviewer 2 Report
In my opinion, the manuscript (MS) requires revisions before reaching a publication. I have several feedback as can be seen on the attached files. I would like to see your response to my suggestions/ questions first before I give my point of you for the next step of publishing process of this MS.

Author Response
Dear Reviewer,
Thank you for your valuable comments regarding manuscript ID: animals-2140071, entitled Effects of Iron Supplementation on Metabolism, Oxidative and Immune Status in Calves receiving whole milk. We appreciate your detailed review and hope that our statements will find your acceptance. English language editing by MDPI (certificate attached, changes shown in blue or red).
Line 80. It could be interesting if 'different sex/ males vs females' is taken into account during the statistical analyses to see the affect of Fe treatment on different sex of calves as well.
AU: Yes, we agree that the effect of treatment would be interesting according to calves’ sex. On the day of supplementation start (7th day of life) blood parameters (iron metabolism) were balanced between male and female. Due to small abundance of animals within gender, we did not divide groups into subgroups. The calves were divided into: control group (CON), fed with full milk (n = 10) and experimental group (MFe), receiving a PIC additive in milk at 16 g/head/day (n = 10); 50% females and 50% males in each group, what make us 5 individuals in subgroup. This would be too small number for statistical analyses. Due to the practical approach (impact by low Fe concentration in blood) we decided to consider the supplement in general.
There is, however, no feed intake data available in this study. ADG is closely affected with the feed intake. Did Fe treatment increase/reduce the feed intake? If the Control group of calves consumed full milk and feed granule higher than those treated by Fe, it was likely the Control group had better performances.
AU: There were no statistically significant differences in milk intake. In general, each calf drank 6 liters of milk per day in both the control and experimental group. Starter feed and hay intake were monitored individually. The use of PIC resulted in an increase in feed intake, especially hay but these data were not statistically significant between the groups. The estimation of hay intake could be inaccurate, which is why we present hard data regarding e.g. ADG.
A sentence has been added:
Line 89-90: The calves had access to clean, fresh water, hay, and starter ad libitum.
16 g/head/day?
AU: Line 85. Done according to the Reviewer’s suggestion.
CR = Average daily gain (ADG, kg/head/day) divided by Dry Matter Intake (DMI, kg/head/day)
How did the DMI from full milk and concentrate granule was measured in individual calf?
FCR = Average daily gain (ADG, kg/head/day) divided by Dry Matter Intake (DMI, kg/head/day)
How did the DMI from full milk and concentrate granule was measured in individual calf?
AU: DM in concentrate granule was determined in laboratory of the Department of Animal Nutrition and Feed Science (according to AOAC 2005 procedures 934.01). The ratio of sample after drying to before drying determined dry matter (DM) of feeds. Calves received a constant amount of DM from whole milk, differences occurred in DM intake from starter feed and hay. Dry matter in whole milk was determined using a drying scale with a properly deduced program for liquid substances (120 °C for 13 min, drying scale Radwag MA.X2, auto program 3).
How did the DMI from full milk and concentrate granule was measured in individual calf?
AU: The starter and hay were offered daily at 8 a.m. and the refusals were weighed daily before preparation of solid feed throughout the experimental period (at 7 a.m.). starter were collected weekly and pooled for analyses. Generally, calves drank a constant amount of milk per day.

Reviewer 3 Report
The article entitled (Effects of Iron Supplementation on Metabolism, Oxidative and Immune Status in Calves receiving whole milk). Authors take inconsideration important point for feeding calves that have problems in iron metabolism in early period of life. But i have many general comments must be improved before being considered for publication.
The manuscript needs intensive languish editing by an expert organization
Manuscript lacks of abbreviations definitions in many parts and sections, also in group definitions.
Here the design of manuscript is very simple and there is no need for analysis by general linear model. The two different groups must be analyzed with T-independent test to be more concise.
Tables lacks of definitions in footnotes
Really manuscript bad organized and not reviewed perfectly from the authors before submission
Author Response
Dear Reviewer,
Thank you for your valuable comments regarding manuscript ID: animals-2140071, entitled Effects of Iron Supplementation on Metabolism, Oxidative and Immune Status in Calves receiving whole milk. We appreciate your detailed review and hope that our statements will find your acceptance. English language editing by MDPI (certificate attached).
The manuscript needs intensive languish editing by an expert organization.
AU: English language editing by MDPI native spikers (changes shown in blue or red).
Manuscript lacks of abbreviations definitions in many parts and sections, also in group definitions.
AU: Done according to the Reviewer’s suggestion.
Abbreviation are expanded when first mentioned in the text. For the rest of text, only abbreviations are used. This form is more transparent in our opinion.
Here the design of manuscript is very simple and there is no need for analysis by general linear model. The two different groups must be analyzed with T-independent test to be more concise.
AU: We agree that it was possible to specify it for individual days between individual dates of the tests. Giving the significance of statistical differences with the means, taking into account repeated measures for given model. Similary model were used also by Hill et al. 2010 (J. Dairy Sci. 93:1105-1115), Soberon et al. 2012 (J. Dairy Sci. 95:783-793).
Tables lacks of definitions in footnotes
AU: Done according to the Reviewer’s suggestion. The tables contain explanations of the abbreviations in footnotes of the tables 1 to 5.

Round 2
Reviewer 3 Report
Thanks for response